# The Effects of TiO₂ Diffuser-Loaded Encapsulation on Corrected Color Temperature Uniformity of Remote Phosphor White LEDs

**Yung-Fang Chou** [1] , **Chi-Feng Chen** [1,*], **Shang-Ping Ying** [2] **and Yun-Ying Yeh** [1]

1   Department of Mechanical Engineering, National Central University, Taoyuan City 32001, Taiwan;
    bill3086@yahoo.com.tw (Y.-F.C.); j9038592@ms38.hinet.net (Y.-Y.Y.)
2   Department of Optoelectronic System Engineering, Minghsin University of Science & Technology,
    Hsinchu 30401, Taiwan; sbying@must.edu.tw
*   Correspondence: ccf@cc.ncu.edu.tw; Tel.: +886-3-426-7308

**Abstract:** With the development of high-efficiency and high-power LEDs, they have become the most energy-efficient and environmentally friendly artificial light source. Phosphor-converted white LEDs are currently mainstream in the market. The remote phosphor is an effective way to enhance the conversion efficiency and lifetime of phosphor-converted LEDs. For applications of high-quality lighting and LCD backlights, the uniformity of angular correlated color temperature (CCT) is very important. This report explored a remote phosphor white LED with low angular CCT variance and high luminous efficiency by using TiO₂ diffuser-loaded encapsulation. Experimental results revealed that for the TiO₂ diffuser-loaded encapsulation remote phosphor white LED, the angular color uniformity could be improved by 31.82% and the luminous flux by 8.65%. Moreover, the mean CCTs of the TiO₂ diffuser-loaded encapsulation and non-diffuser remote phosphor white LEDs were similar at a driving current of 350 mA. Finally, we showed that incorporating the TiO₂ diffuser into the phosphor layer of the remote phosphor white LEDs, does not influence the reliability of the LED.

**Keywords:** TiO₂ diffuser; corrected color temperature (CCT); remote phosphor white LED

## 1. Introduction

In recent years, white LEDs based on phosphor conversion have constituted a considerable proportion of the LED market because of their many advantages, such as long lifetime, high reliability, environmental protection, safety, and multiple applications [1]. To meet the requirements of sustainable development, continual enhancement of the luminous efficiency of LEDs is crucial. A remote phosphor package that separates the phosphor layer from the chip to suppress reabsorption is an efficient method for enhancing luminous efficiency. Remote phosphor white LEDs with the LED chip and phosphor layer at the bottom and top of a reflector cup were investigated [2]. The experimental results showed that the backscattered photons can be extracted and the efficacy can be significantly increased by separating the phosphor layer from the chip. For white LEDs with a diffuse reflector cup, the numerical and experimental results showed that light extraction efficiencies were enhanced by 75% and 15.4%, respectively [3]. The optimization investigation on packaging configuration for remote phosphor white LEDs comprising the LED chip, the phosphor layer, a diffuse reflector cup, and a hemispherically shaped encapsulation is presented. The numerical and experimental results show that luminous efficiencies can be enhanced by up to 50% and 15.4% over conventional packages, respectively [4]. A remote phosphor packaging was numerically and experimentally investigated. The results prove that this separate packaging can prevent the heat of the LED chip from transferring to the phosphor layer and improve the luminous efficiency and color characteristic stability of the white LED [5].

However, in typical remote phosphor white LEDs, the phosphor-scattered blue light and phosphor-emitted yellow light have different radiant intensity distributions. Therefore, the angular color distribution becomes non-uniform, and a yellow ring phenomenon appears in the illuminating plane. Several research studies have focused on the improvement of angular color distribution for remote phosphor white LEDs. A simulation analysis of color distribution for white LEDs with five phosphor packaging methods was presented [6]. The results showed that, for the effects of angular color distribution, the most important issue was the packaging method, followed by the reflector cup, and the minor was the separation distance between the phosphor layer and the LED chip. Among these methods, the angular color uniformity for the types with a plane phosphor layer was poor and that of the types with a convex phosphor layer was better. Based on ray-tracing simulations, Hong et al. showed that the uniformity of angular correlated color temperature (CCT) was enhanced by optimizing the color conversion element geometries and material compositions [7]. Huang et al. proposed that the remote phosphor of fully patterned sapphire provides a much better CCT uniformity than that obtained in the cases of planar sapphire and partially patterned sapphire [8]. Chen et al. demonstrated that adding a silicone layer with $ZrO_2$ particles onto the surface of the remote phosphor layer could enhance the capability of light scattering [9]. The structure enhanced the uniformity of angular CCT and increased the luminous efficiency by a small amount. Kuo et al. proposed a patterned remote phosphor structure in which not spraying phosphor on a window region could improve CCT uniformity and luminous efficiency, as compared with the conventional remote phosphor coating structure [10]. Ding et al. proposed the addition of a micro-patterned array optical film into the packaging configuration for remote phosphor white LEDs [11]. The numerical and experimental results showed that this optical film could improve the uniformity of angular CCT. Yang et al. proposed the effects of melamine formaldehyde resin and $CaCO_3$ diffuser-loaded encapsulation on the correlated color temperature uniformity of phosphor-converted LEDs [12]. The experimental results showed that this method could enhance the angular color uniformity of LEDs. Lai et al. proposed the addition of $SiO_2$ scattering particles into the phosphor layer of a multi-chip white light LED. The simulation showed that $SiO_2$ particles could significantly enhance the CCT uniformity and luminous flux [13]. While these methods of improvement have been confirmed by achieving high CCT uniformity, they also engender significant fabrication difficulties for mass production.

It is necessary to find a remote phosphor packaging method that not only realizes high CCT uniformity but is also cheap and easy to fabricate. For this reason, we are interested in a more convenient method of manufacturing for remote phosphor packaging, for example, nanoparticle diffuser-loaded encapsulation methods— encapsulation containing nanoparticle diffusers cause light to reflect, refract, and scatter, thereby randomizing the propagation direction and isotropizing the far-field distribution. Minh et al. proposed an innovative method consisting of mixing $SiO_2$ and $Sr_2Si_5N_8$:$Eu^{2+}$ phosphor particles into the YAG:Ce phosphor of remote phosphor LEDs to improve the color uniformity and color rending index (CRI) [14]. By fixing the concentration of $SiO_2$ particles at 5%, the simulation results showed that color uniformity and CRI significantly improved with a rising concentration of $Sr_2Si_5N_8$:$Eu^{2+}$ phosphor particles. Yuce et al. investigated the luminous efficiency of the generated white light from Polydimethylsiloxane (PDMS)/phosphor composites through employing various glassy scattering particles, such as glass beads, glass bubbles, or $SiO_2$ nanoparticles [15]. It was demonstrated that the luminous efficiency of remote phosphor LEDs could be improved and controlled by the morphology of the scatter, the volume fraction of the scatters, and the thickness of the PDMS/YAG:$Ce^{3+}$ composite samples. In this study, we used $TiO_2$ diffuser-loaded encapsulation to improve the CCT uniformity and decrease the CCT variance in remote phosphor white LEDs. The $TiO_2$ diffuser can provide superior scattering capability for light. The effects of different positions and concentrations of diffusers on the spatial CCT distribution, light extraction efficiency, and CCT of remote phosphor white LEDs were investigated to find the optimized solution.

## 2. Experiment

In this study, rutile crystalline-form $TiO_2$ (Shun-Yi Inc., Taichung, Taiwan) was chosen to be the diffuser loaded in the remote phosphor white LED. The refractive index and density of the $TiO_2$ diffuser were determined to be 2.9 and 4.26 $g/cm^3$, respectively. Qi proposed that the particle size and particle size distribution of the $TiO_2$ diffuser influence the results of optical scattering [16]. Therefore, before starting our experiment, we used a laser scattering particle size distribution analyzer (type: HORIBA, LA-950, Trendtop scientific corp., Taipei, Taiwan) to measure the particle size and particle size distribution of the $TiO_2$ diffuser. The mean size of the $TiO_2$ diffuser was measured to be 39.8 nm. The particle size and undersize distribution of the $TiO_2$ diffusers are shown in Figure 1. The mean particle size of the $TiO_2$ diffusers was much less than one-tenth of the light wavelength in the visible band, where Rayleigh scattering is the dominant scattering mechanism [17].

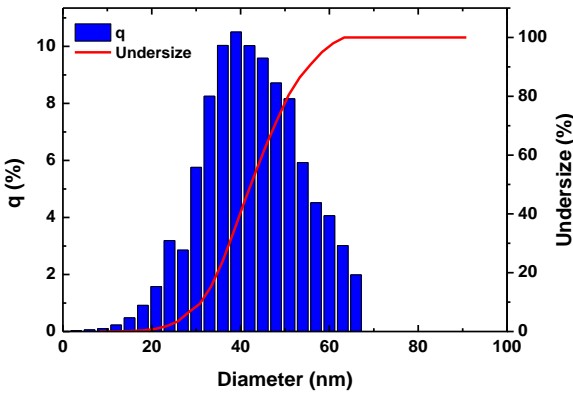

**Figure 1.** Schematic of the particle size distribution of the $TiO_2$ diffuser.

Samples with a $TiO_2$ diffuser in the remote phosphor LEDs were fabricated with the following steps: A blue LED chip (type: ES-CEBLV40A-M, Epistar Inc., Hsinchu, Taiwan) with a size of $40 \times 40$ mil and a peak emission wavelength of 460 nm was placed in a plastic lead-frame package (type: TTI-5074, I-Chiun Inc., Taipei, Taiwan). In order to meet statistical reliability, we tested 10 samples for each different experiment parameter and averaged the measurement results of the samples before discussing the experimental results in this study.

### 2.1. Diffuser in the Encapsulation Layer

To evaluate the effects of encapsulation on the CCT uniformity of the $TiO_2$ diffuser in the encapsulation layer of the remote phosphor LEDs, the encapsulation layer was mixed with $TiO_2$ diffuser contents of 0, 0.2, 0.4, 0.6, and 0.8 wt.%. The schematic diagram of the $TiO_2$ diffuser in the encapsulation layer of the remote phosphor LED is shown in Figure 2a. The total weight of the encapsulation layer was the sum of the weights of the silicone resin (type: Dow Corning OE-6370 HF) and the $TiO_2$ diffuser. A uniform mixture of the silicone resin and $TiO_2$ diffuser was injected into the lead frame through the dispenser, where the dispense pressure and dispensing time were set at 30 psi and 1.8 s. Subsequently, the LED samples were baked in a vacuum oven at 50 °C for 60 min and then at 150 °C for 90 min. Approximately 17 wt.% phosphor (type: Intermatix YAG-04, Intermatix Company, Fremont, CA, USA) mixed with silicone resin was then coated on the phosphor layer through the dispenser, where the dispense pressure and dispensing time were set at 30 psi and 0.8 s. Finally, the LED samples were baked in a vacuum oven at 50 °C for 60 min and then at 150 °C for 90 min.

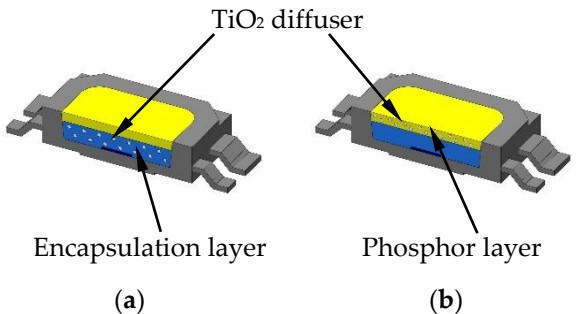

TiO₂ diffuser

Encapsulation layer      Phosphor layer

(**a**)          (**b**)

**Figure 2.** Schematic diagrams of the TiO$_2$ diffuser in the (**a**) encapsulation layer and (**b**) phosphor layer of the remote phosphor LEDs.

### 2.2. Diffuser in the Phosphor Layer

To evaluate the effects on the CCT uniformity of the TiO$_2$ diffuser in the phosphor layer of the remote phosphor LEDs, the phosphor layer was mixed with TiO$_2$ diffuser contents of 0, 0.2, 0.4, 0.6, and 0.8 wt.%. The schematic diagram of the TiO$_2$ diffuser in the phosphor layer of the remote phosphor LED is shown in Figure 2b. The total weight of the phosphor layer was the sum of the silicone resin, phosphor, and TiO$_2$ diffuser in the remote phosphor LEDs. The silicone resin only was first injected into the lead frame through the dispenser, where the dispense pressure and dispensing time were set at 30 psi and 1.8 s. Subsequently, the sample was baked in a vacuum oven at 50 °C for 60 min and then at 150 °C for 90 min. A uniform mixture of approximately 17 wt.% phosphor, TiO$_2$ diffuser, and silicone resin was coated on the encapsulation layer through the dispenser, where the dispense pressure and dispensing time were set at 30 psi and 0.8 s. Finally, the LED samples were baked in a vacuum oven at 50 °C for 60 min and then at 150 °C for 90 min.

### 2.3. Optical Properties Measurement

The optical properties of the manufactured remote phosphor white LEDs were measured at 350 mA at a viewing angle ranging from −70° to 70°. A viewing angle range from −90° to 90° was not used for evaluation because luminous intensity is very weak around this range. Luminous flux and CCT at 350 mA were measured with the LED Test and Measurement System (CAS 140CT, Instrument Systems). CCT by angle, from −70° to 70° with a 5° interval, was measured at 350 mA with the Light Intensity Distribution Meter (type: LID-200, OPTIMUM Corp., Hsinchu, Taiwan). The spatial CCT P-V deviation, which indicates the difference between the maximum and minimum spatial CCT ranging from −70° to 70°, served as the index of angular color uniformity.

### 3. Discussion

The changes in CCT and CCT P-V deviation by TiO$_2$ diffuser positions and diffuser contents in the examined remote phosphor LEDs are shown in Figure 3a,b. Increasing the TiO$_2$ diffuser content was associated with a reduction in CCT and CCT P-V deviation. The higher the CCT distribution, the larger the CCT P-V deviation.

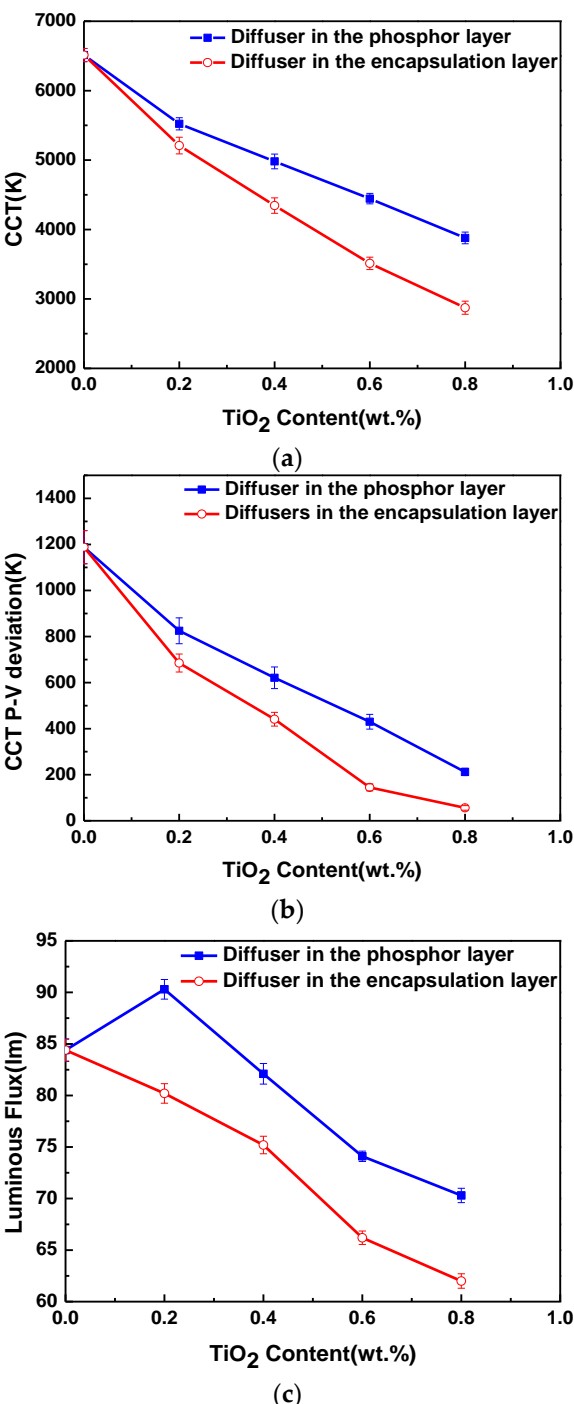

**Figure 3.** (**a**) The correlated color temperature (CCT), (**b**) the CCT P-V deviation, and (**c**) the luminous flux of the remote phosphor white LEDs for different TiO$_2$ contents.

### 3.1. Results of Different Diffuser Positions and Diffuser Contents in the Remote Phosphor LEDs

As illustrated in Figure 3a, the CCT of the remote phosphor white LED was 6513 K when the TiO$_2$ diffuser content was 0; the CCT decreased to 2874 K when the diffuser content in the encapsulation layer was 0.8 wt.% and to 3869 K when the content in the phosphor layer was 0.8 wt.%. These findings can be explained as follows—the CCT of remote phosphor LEDs is determined by the ratio of the intensity of yellow light to that of blue light. The blue light emitted from LED chips excites phosphors to emit yellow light, and the Rayleigh scattering effect of the TiO$_2$ diffuser increases the amount of blue light and thus, leads to more yellow light. Hence, the intensity of the blue light decreases and that

of the yellow light increases, thus reducing the CCT of the LEDs regardless of the diffuser position. This study determined that at a constant diffuser content, the CCT was lower in the case that involved diffuser in the encapsulation layer than in the case that involved a diffuser in the phosphor layer. This was engendered by the larger scattering volume of the diffusers in the encapsulation layer, leading to a greater Rayleigh scattering effect.

Figure 3b demonstrates that adding the $TiO_2$ diffuser to the remote phosphor LEDs could improve the color temperature uniformity across the package. The diffusers could significantly reduce the CCT variance regardless of their position. The $TiO_2$ diffuser had more favorable effects in the encapsulation layer than in the phosphor layer, because the encapsulation layer was associated with the lowest CCT P-V deviation across all diffuser contents. At the diffuser content of 0.8 wt.%, the reduction rate of the CCT P-V deviation was approximately 82.15% in the phosphor layer [(1188 – 212)/1188 $\times$ 100% = 82.15%], whereas that in the encapsulation layer was nearly 95.29% [(1188 – 56)/1188 $\times$ 100% = 95.29%]. Thus, the diffusers in the encapsulation layer were associated with a better uniformity of CCT.

The measurement results in Figure 3c reveal that the $TiO_2$ diffuser content influenced the lumen output of the remote phosphor LEDs. The $TiO_2$ diffuser increased the light output of a remote phosphor LED by more than 6.99% [(90.3 – 84.4)/84.4 $\times$ 100% = 6.99%] when the inert diffuser content was 0.2 wt.% of the phosphor layer. In addition, the lumen output slightly increased because the effect of diffuser scattering could suppress the optical trapped modes with the rays inside the remote phosphor white LEDs having more than ten reflections from the Ag reflector cup. This was identified to be a significant light loss mechanism. Diffuser scattering led to the escape of light into free space, thus increasing the light extraction efficiency [18]. Then, the $TiO_2$ diffuser content increased and the lumen output decreased due to the excessive backward scattering and absorption of light by the encapsulant. As the $TiO_2$ diffuser content increased, the changes in the luminous flux increased, with the $TiO_2$ diffuser in the phosphor layer exhibiting a higher luminous flux than that in the encapsulation layer. This could be because the phosphor layer is thinner and can thus absorb less light. For diffuser-loaded encapsulants in a silicon layer, the reduction in transmitted light intensity due to light scattering can be described by [19,20]:

$$\frac{I}{I_0} = exp\left[\frac{-3Lxr^3}{4\lambda^4}\left(\frac{n_d}{n_s} - 1\right)\right]$$
(1)

where $I$ is the intensity of light passing through the sample, $I_0$ is the intensity of light that would pass through the sample without scattering, $x$ is the optical path length, $L$ is the diffuser volume loading fraction, $r$ is the diffuser radius, $\lambda$ is the light wavelength, and $n_d$ and $n_s$ are the refractive indices of the diffuser and silicon, respectively. Equation (1) also proves that the light intensity decreases as L (i.e., $TiO_2$ diffuser content) and $x$ (i.e., the thickness of the encapsulation layer or phosphor layer) increase.

### 3.2. Optimum Design of the Remote Phosphor LED

Because $TiO_2$ diffuser-loaded encapsulation in the remote phosphor LEDs generated not only light-scattering effects but also light absorption to reduce luminous flux, we sought the optimum conditions for enhancing the angular CCT uniformity and maintaining a higher level of luminous output. Accordingly, the $TiO_2$ diffuser content of 0.2 wt.% in the phosphor layer was determined to be an adequate condition for obtaining a lower CCT P-V deviation and a higher luminous flux. However, this would concurrently reduce the CCT from 6513 K to 5524 K of remote phosphor white LEDs. Therefore, to preserve the CCT in the range of 6500 $\pm$ 100 K, the phosphor content in the phosphor layer of the remote phosphor white LEDs must be adjusted. After the experiments, we adjusted the phosphor content from the original 17.0 wt.% to 16.0 wt.% and the CCT of the remote phosphor LED was changed to 6516 K. Figure 4a shows the changes in the CCT resulting from changes in the content of the silicon during the mixing of the phosphor with the $TiO_2$ diffuser content of 0.2 wt.%. Figure 4b shows the electroluminescence spectra of the $TiO_2$ diffuser-loaded encapsulation remote phosphor

LED at a driving current of 350 mA. The relative spectrum has two specific peaks, centered at 460 and 570 nm. The results in this study utilized a blue LED chip and phosphor to produce white light.

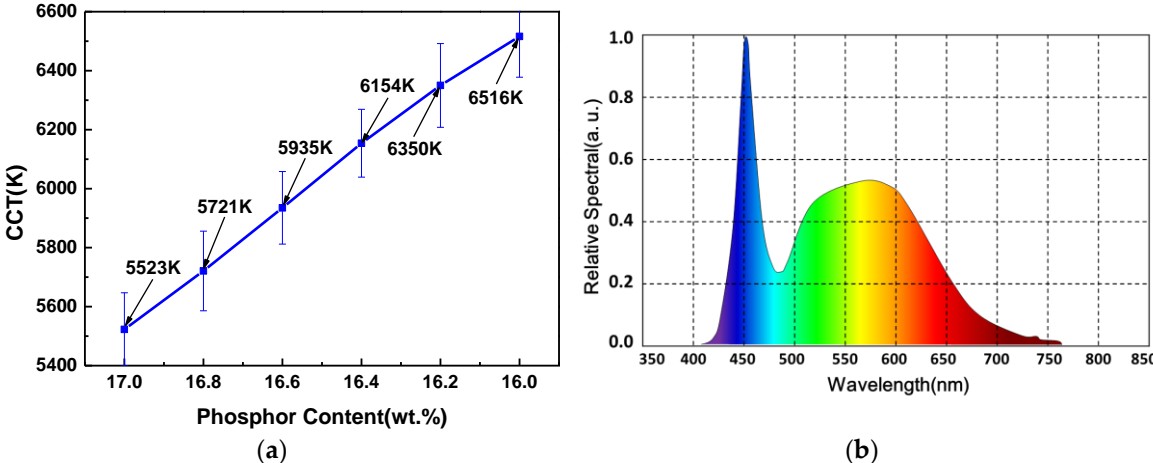

**Figure 4.** (**a**) Dependence of CCT on the phosphor content for the remote phosphor white LEDs with the TiO$_2$ diffuser content of 0.2 wt.% in the phosphor layer, (**b**) Electroluminescence spectra of the TiO$_2$ diffuser-loaded encapsulation remote phosphor LED at a driving current of 350 mA.

The angular CCT distribution of the non-diffuser and TiO$_2$ diffuser-loaded encapsulation remote phosphor white LEDs was measured, as illustrated in Figure 5. It is clear that the angular CCT deviation of the non-diffuser remote phosphor LED was improved by 31.82% [(1188 − 810)/1188 × 100% = 31.82%] by the TiO$_2$ diffuser-loaded encapsulation remote phosphor LED in the range of −70° to 70°. The non-diffuser remote phosphor LED had a larger angular CCT deviation, which was due to the longer optical path for the high angles, where the blue photons can be converted more easily by the phosphor particles than the normal directions, limiting the extraction of blue light at this angle and consequently leading to ineffective color mixing of blue and yellow light. The TiO$_2$ diffuser-loaded encapsulation remote phosphor LED was observed to have a smoother angular CCT distribution, because the TiO$_2$ diffuser could provide an effective scattering capability to improve the ratio of yellow to blue light at large angles.

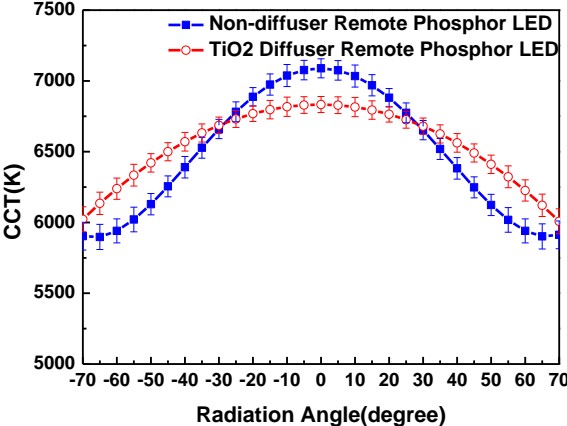

**Figure 5.** Angular CCT distribution of non-diffuser and TiO$_2$ diffuser-loaded encapsulation remote phosphor LEDs.

The CCTs for both remote phosphor LEDs were almost the same (6513 and 6516 K) at a driving current of 350 mA. To further understand the optical characteristics of the TiO$_2$ diffuser-loaded encapsulation remote phosphor LED, we measured the current-dependent luminous flux and

calculated the normalization of luminous efficacy. As presented in Figure 6a, the luminous flux of the TiO$_2$ diffuser-loaded encapsulation remote phosphor LED slightly increased by 8.65% [(91.7 - 84.4)/84.4 × 100% = 8.65%], at a driving current of 350 mA, compared with the non-diffuser remote phosphor LED. As shown in Figure 6b, the normalization of luminous efficacy of the TiO$_2$ diffuser-loaded encapsulation remote phosphor LED slightly increased by 7.81% [0.801 − 0.743)/0.743 × 100% = 7.81%], at a driving current of 350 mA, compared with the non-diffuser remote phosphor LED. The slopes of the normalization efficacy curves have similar trends for the two samples, this is because they use the same LED chip and phosphor.

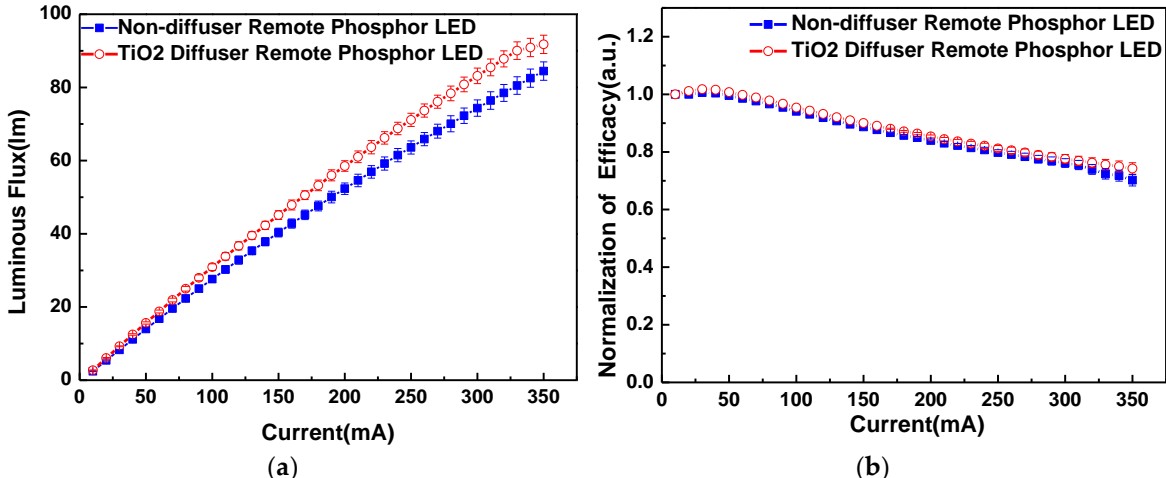

**Figure 6.** (**a**) Variations in output luminous flux and (**b**) normalization of efficacy as a function of injection current in non-diffuser and TiO$_2$ diffuser-loaded encapsulation remote phosphor LEDs.

A comparison of the non-diffuser and TiO$_2$ diffuser-loaded encapsulation remote phosphor white LEDs, in terms of light projection uniformity, is shown in Figure 7a,b. In the light projection of the non-diffuser remote phosphor white LED, an obvious non-uniform phenomenon could be observed on the boundary of the light spot. In the light projection of the TiO$_2$ diffuser-loaded encapsulation remote phosphor white LED, the uniformity of the light spot improved. With the scattering characteristics of the TiO$_2$ diffuser, the blue light and yellow light could be distributed uniformly.

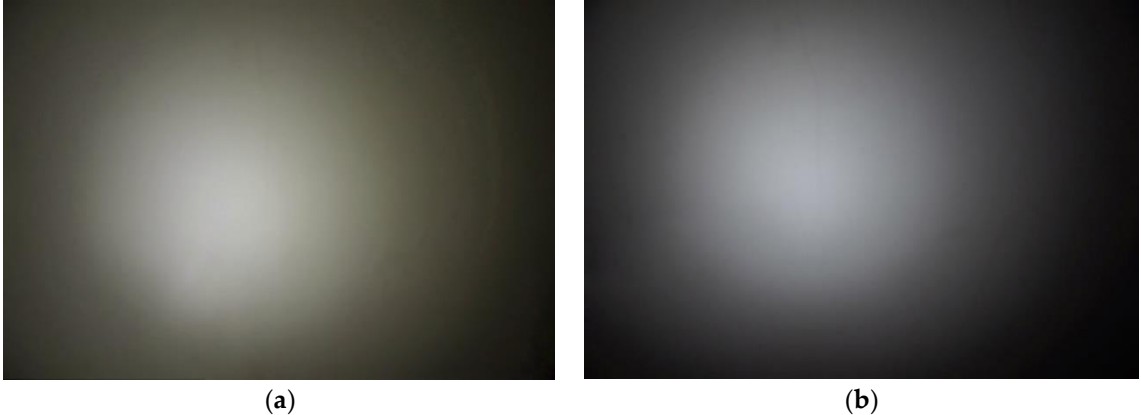

**Figure 7.** Light projection at 350 mA for (**a**) non-diffuser and (**b**) TiO$_2$ diffuser-loaded encapsulation remote phosphor LEDs.

### 3.3. Reliability Analysis of the TiO$_2$ Diffuser-Loaded Encapsulation Remote Phosphor LED

A reliability analysis was conducted with the following test conditions: temperature of 60 °C; humidity of 90 RH (%); and injection current of 350 mA. Both remote phosphor white LED types were

measured at room temperature. Normalization of luminous flux curves varied with time, as illustrated in Figure 8. There was a steep change in the slope of the lumen deterioration curve of the LED samples between 0 and 1500 h, but they started to flatten out after 1500 h. Both LED samples had similar slope values of decreasing luminous flux. Normalization of the luminous flux curves showed that incorporating the $TiO_2$ diffuser into the phosphor layer of the remote phosphor white LED did not influence the reliability of the LED.

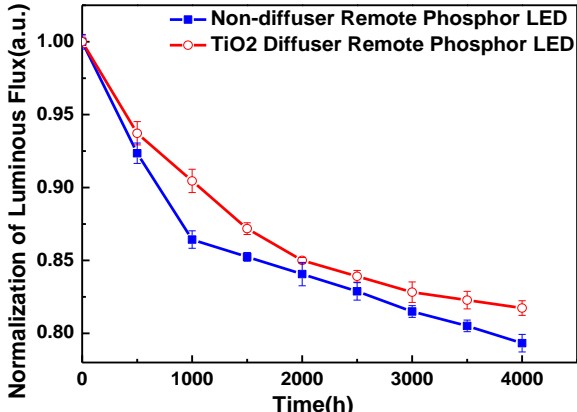

**Figure 8.** Reliability analysis of non-diffuser and $TiO_2$ diffuser-loaded encapsulation remote phosphor LEDs (350 mA injection current).

## 4. Conclusions

In this study, the utility of low-cost and controllable $TiO_2$ diffuser-loaded encapsulation provided an effective method for improving the angular color uniformity of remote phosphor white LEDs. An inert $TiO_2$ diffuser content of 0.2 wt.% and a phosphor content of 16.0 wt.% in the phosphor layer were determined to be the optimum conditions for obtaining low angular CCT variance, high luminous efficiency, and maintaining the CCT of white LEDs, simultaneously. We found that the angular color uniformity could be improved by 31.82% in the $TiO_2$ diffuser-loaded encapsulation remote phosphor LED, compared with the non-diffuser remote phosphor LED. At a driving current of 350 mA, the luminous flux of the $TiO_2$ diffuser-loaded encapsulation remote phosphor LED was increased by 8.65% relative to the non-diffuser remote phosphor LED. Finally, we showed that incorporating the $TiO_2$ diffuser into the phosphor layer of the remote phosphor white LED did not influence the reliability of the LED.

**Author Contributions:** All the authors participated in the design of experiments, analysis of data and results, and writing of the paper.

**Funding:** This research received no external funding.

**Acknowledgments:** We thank Dr. M. D. Lin for providing some measurement equipment

**Conflicts of Interest:** The authors declare no conflicts of interest.

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
