# Peer review of "The Effects of TiO2 Diffuser-Loaded Encapsulation on Corrected Color Temperature Uniformity of Remote Phosphor White LEDs"

_applsci, doi:10.3390/app9040675_

Round 1
Reviewer 1 Report
A reply to the report of “applsci-418498”
This is comments on the article entitled by “The effects of TiO2 diffuser-loaded encapsulation on corrected color temperature uniformity of remote phosphor white LEDs” by Chou et al.
This article has studied that a remote phosphor white LED with TiO2 diffuse-loaded encapsulation affected the low angular correlated color temperature (CCT) variance and high luminous efficiency. The authors stated that the CCT in the range 6500K depends on change of phosphor content 17% to 16% with the fixed 0.2wt % of TiO2 in phosphor layer of remote white LED. As I think that this manuscript is not appropriate for publication in Applied Science, I consider it possible to accept this manuscript after major revision.
1. Authors should modify the luminous flux versus time by normalization of luminous flux for reliability analysis as shown Figure 8. I think that both samples have the same slope values of decreased luminous flux. Additionally, I wonder if authors foresee the life-time of proposed remote phosphor LED by measured accelerated-life test for reliability analysis.
2. Authors have referred “The higher light extraction efficiency can reduce the heat” in 273 lines, 8 pages. I think that the higher light extraction efficiency of remote phosphor LED occurred by the inserted TiO2 particle in phosphor layer as compared to conventional remote LED. This sentence is insufficient meaning in article. Authors should show more results or modify the referred sentence.
3. Authors have showed the luminous efficacy in Figure 6(b). However, the slope of efficacy have similar trend between two samples. Therefore, Authors should change the normalization of calculated luminous efficacy versus injection current graph, and mention the scientific results.
4. Authors should specify (a) and (b) in the Figure 2.
5. Authors could display the electroluminescence data of both encapsulation layer and phosphor layer of the remote phosphor LEDs.
6. Authors should modify “The introduction should briefly ~~ on references” in 52 – 59 lines, 2 pages, Introduction part.
Author Response
Please see the attach files.

Reviewer 2 Report
The authors report the effect of inclusion of nano-particles in the remote phosphor-based white LEDs and suggest that this design could improve efficacy, angular color uniformity, and reliability. The experimental results of this study partially support the authors’ conclusions, but there are several points which need to be clarified in the revised version.
- In Section “2. Theory”, the authors wrote that the Rayleigh scattering is the main light scattering mechanism of TiO2 particles because of their small sizes. As the Eq.(1) demonstrates, the most important characteristic of the Rayleigh scattering is its wavelength dependence. As the wavelength decreases, the scattering ability increases significantly. However, the authors did not mention anything about the wavelength dependence of the Rayleigh scattering in their discussions. I was wondering whether the authors really did not consider the effect of the wavelength dependence on their observations.
- line 91: the dimensions of the LED chip and the package should be given in exact numbers. “square mils” should be corrected.
- Line 152, line 159: the result in Fig. 3(b) cannot be used to mention either the spatial uniformity or the angular uniformity.
- line 182: the authors need to describe in more detail about the “optical trapped modes”.
- lines 222~225: the CCT deviation was explained by the “trapped and reflected” blue light. I think one more (and probably more important) reason is the longer optical path for the high angles, where the blue photons can be converted more easily by the phosphor particles, than the normal directions (shorter optical paths).
- The unit of Fig. 6(b) is wrong.
- Fig. 7 should be in color instead of gray levels.
- The authors relate the result in Fig. 8 with significant improvement of the reliability. However, this figure shows that the lumen decrement is similar in both cases (~15% vs ~20% ??). Moreover, the number of tested samples is not mentioned, which means the results are not statistically reliable. Especially, the difference in the efficacy between the two samples operated at 350 mA are very small as the Fig. 6(b) shows. Therefore, it is difficult to explain the reliability improvement (if it exists) to the different generation of heat.
- The present version is incomplete. The authors did not trim the manuscript, for example, lines 52-29 (which is very strange). There are many important papers about the remote phosphor-based white LEDs, but the authors’ references cited only few of them. References on the others’ works need to be included and explained in the introduction.
Author Response
Please see the attach file.

Reviewer 3 Report
The manuscript describes the effect of TiO2 on corrected color temperature uniformity of remote phosphor white LEDs. Experimental results revealed that for the novel remote phosphor white LED, the angular color uniformity could be improved by 31.82% and the luminous flux by 8.65%. This is a very good piece of work and well presented. This report is recommended to publish in applied sciences after the following minor revisions.

Author Response
Please see the attach file.

Round 2
Reviewer 1 Report
A reply to the report of “applsci-418498”
This is comments on the article entitled by “The effects of TiO2 diffuser-loaded encapsulation on corrected color temperature uniformity of remote phosphor white LEDs” by Chou et al.
The authors have revised the article according to comments. However, “novel” word in article is not appropriate as compared to diverse research papers and introduction part in the revised manuscript has wrong reference number. Authors should peruse and modify the article. Therefore, I consider it possible to accept this manuscript after revision.
Author Response
Please see the attach file.

Reviewer 2 Report
The manuscript was improved based on the reviewers' comments.
Some more recommendations for further improvements are as follows.
(1) The authors removed the "Theory" part in the revised version. However, I think Rayleigh scattering plays some role in interpreting the experimental results because the particle sizes of TiO2 correspond to the Rayleigh scattering region. For example, the TiO2 particles scatter more blue light and less yellow light, which means blue photons cannot escape well from either the encapsulation layer or the phosphor layer. If the authors mention only "scattering effect", it equally applies to all photons (blue and yellow). However, to mention "Rayleigh scattering effect" suggests that the scattering effect is more severe for blue photons. I think this is one of the reasons for the observation summarized in Figure 3.
(2) The authors mentioned that they tested 10 samples. They should include the standard deviation for all measured properties presented in the figures.
(3) When describing the result in Fig. 8, "light attenuation" is not appropriate because attenuation is used from different meaning in optics. Instead "lumen deterioration curve" or "lumen maintenance curve" is more conventional expression. In addition, English still needs improvement. I can find many grammatical errors. I recommend the authors to get some consult from native speakers.
(4) To cite other researchers' work is an essential and prerequisite part in academic papers. The reference list is still incomplete not including more recent papers. For example, very similar approaches were already published recently (Optics Letters Vol. 44, Issue 3, pp. 479-482 (2019), Materials Science-Poland 36, 370 (2018)) where scattering particles were embedded in the design. Color uniformity or improved reliability was also reported in recent publications including spherical-shaped remote phosphors (Materials Letters Volume 227, 104-107 (2018), Science of Advanced Materials, Volume 8, 342 (2016), just to name a few). I cannot understand why the authors did not cite any paper published between years 2016-2019 except only one. Especially, the comparison between the present work and the two former papers mentioned above (2019 and 2018) seems necessary.
Once these issues are duly addressed in the next revision, it may be accepted for publication in this journal.
Author Response
Please see the attach file.
